# Immune Modulation Using Extracellular Vesicles Encapsulated with MicroRNAs as Novel Drug Delivery Systems

**DOI:** 10.3390/ijms23105658

**Published:** 2022-05-18

**Authors:** Yasunari Matsuzaka, Ryu Yashiro

**Affiliations:** 1Division of Molecular and Medical Genetics, Center for Gene and Cell Therapy, The Institute of Medical Science, University of Tokyo, Minato-ku 108-8639, Tokyo, Japan; 2Administrative Section of Radiation Protection, National Institute of Neuroscience, National Center of Neurology and Psychiatry, Kodaira 187-8551, Tokyo, Japan; ryuy@ncnp.go.jp or

**Keywords:** exosomes, extracellular vesicles, immune regulation via miRNAs, immune modulation, immune tolerance, macrophages

## Abstract

Self-tolerance involves protection from self-reactive B and T cells via negative selection during differentiation, programmed cell death, and inhibition of regulatory T cells. The breakdown of immune tolerance triggers various autoimmune diseases, owing to a lack of distinction between self-antigens and non-self-antigens. Exosomes are non-particles that are approximately 50–130 nm in diameter. Extracellular vesicles can be used for in vivo cell-free transmission to enable intracellular delivery of proteins and nucleic acids, including microRNAs (miRNAs). miRNAs encapsulated in exosomes can regulate the molecular pathways involved in the immune response through post-transcriptional regulation. Herein, we sought to summarize and review the molecular mechanisms whereby exosomal miRNAs modulate the expression of genes involved in the immune response.

## 1. Introduction

Self-tolerance, a process in which the body protects itself from self-reactive B and T cells via a lack of immune response, includes central and peripheral tolerance [1,2,3]. In central tolerance, B cells originate in the bone marrow and T cells originate in the thymus, and these cells undergo negative selection during differentiation [4,5]. Notably, autoreactive immature cell clones die via programmed cell death [6]. Peripheral tolerance involves the elimination or inactivation of mature autoreactive cells that escaped central tolerance through various mechanisms, such as anergy, where immune cells cannot initiate a complete response to their target; ignorance; programmed cell death; and inhibition of regulatory T cells, a type of T cell that exerts immune tolerance effects via direct contact with target cells or the release of immunosuppressive factors (Figure 1) [7,8,9]. When immune tolerance breaks down and the immune system cannot distinguish between self-antigens and non-self-antigens, normal cells and tissues are attacked, triggering autoimmune diseases, such as systemic lupus erythematosus, systemic scleroderma, polymyositis/dermatomyositis, Sjogren’s syndrome, Behcet’s disease, rheumatoid arthritis, etc. [10,11,12,13]. Continued immune response leads to the development of chronic inflammation. The following mechanisms are involved in the autoimmune process: (1) self-reactive helper T1 (Th1) cells release interferon gamma (IFN-gamma) and interleukin (IL)-17, activating macrophages that release cytokines, such as TNF-α and IL-1, causing local inflammation; (2) self-reactive cytotoxic T cells (Tc) damage different tissues; (3) inappropriate T-cell responses cause autoreactive B-cell clones to proliferate and produce autoantibodies; and (4) autoantibodies activate the complement system to cause inflammation and inhibit signal transduction via hormone and neurotransmitter receptors on the cell surface, causing the binding of antigens in the blood to form useless complexes [14,15,16,17,18]. An important factor in immune tolerance is the maintenance of T cell homeostasis, such as the removal of autoreactive T cells in central tolerance, whose main mechanism is cell death due to apoptosis [19,20]. However, another form of cell death, necroptosis (i.e., programmed necrosis) is induced by TNF-α through a mixed lineage kinase domain-like (MLKL), a substrate of receptor-interacting protein kinase 3 (RIP3), which may also be involved in the removal process [21,22]. The breakdown of immune tolerance, which leads to the induction of autoimmune diseases, is caused by the following: (1) virus infection of tissues, which induces the growth of virus-specific T cells, resulting in anergy exceeding its capacity; (2) an autoantigen with high homology to an epitope of an immune reaction induced by infection or a self-antigen (molecular mimicry); (3) the attack of normal cells by immune cells against tumor cells as they are or will be activated [23,24,25]. In this process, somatic mutations reduce immunogenicity and antigens secreted by tumor cells in the equilibrium phase of proliferation, and the immune response induces cross reactivity with proteins in normal cells [26]. (4) Damage-associated molecular patterns (DAMPs), such as heat shock proteins (HSPs) released by cell stress and damage, activate the immune system, thereby inducing an autoimmune response [27]. (5) Nucleic acids released from cells that have undergone cell death stimulate Toll-like receptors (TLRs) on B cells, inducing the production of autoantibodies [28]. Autoimmune diseases are also caused by the strong association of the human leukocyte antigens, HLA class I and class II alleles, with specific autoimmune diseases, and genetic mutations in CTLA4, a suppressor receptor for T cell responses, and PTPN22, which acts as a negative regulator of T cell receptor signaling [29,30].

## 2. Immune-Cell-Derived Exosome as an Immune Modulator

Cells secrete various protein factors, such as growth factors and cytokines, in response to external stimuli and transduce molecular signals to target cells themselves, locally or throughout the body. In addition to these humoral factors, vesicles comprising biological membranes of phospholipid bilayers (hereinafter referred to as membrane vesicles) are secreted regardless of cell types, such as hematopoietic cells (i.e., platelets, B cells, T cells, dendritic cells, mast cells) and non-hematopoietic cells (i.e., small intestinal epithelial cells, cancer cells, Schwann cells, nerve cells, fat gland cells, sperm, and mammary epithelial cells) [31,32,33]. Specific proteins exist on or in the membranes of membrane vesicles. As for the secretion of membrane vesicles, the first report was shown as a means for reticulocytes to “discard” their membrane proteins, such as transferrin receptors, which are no longer needed during differentiation [34]. However, membrane vesicles secreted from various cells, especially immunocompetent and intestinal epithelial cells, have now been recognized to be involved in antigen presentation and the induction of immune tolerance [35]. In addition to signal transduction, membrane vesicles have been suggested to function as inter-cell carriers for molecules, such as proteins, nucleic acids, or lipids [36]. Membrane vesicles secreted by cells are classified into two types, according to their formation and secretory mechanisms. [37,38].

In recent years, the therapeutic application of various cells that release small (diameter of 30–100 nm) membrane vesicles has received remarkable attention [39]. The density of exosomes, which are one of small membrane vesicles released from the fusion of multivesicular bodies (MVBs) with the plasma membrane, can be distributed in fractions of approximately 1.12 to 1.19 g/mL during ultracentrifugation and exosomes can carry out intercellular communications due to their ability to exchange and transmit molecular information [40,41,42]. Proteins on the surface of cell membranes, such as growth factors, are taken up into the cell by clathrin-, caveolin-, and lipid raft-mediated endocytosis, phagocytosis, micropinocytosis, or membrane fusion [43]. Further, early and late endosomes are formed via endocytic vesicles [44]. Although some proteins are degraded by lysosomes via late endosomes, other proteins migrate into late endosomes with small intraluminal membrane vesicles (ILVs), formed by further inward constriction of the late endosome membrane [45,46]. ILVs are exocytosed via the fusion of MVBs containing some ILVs with the cell membrane as exosomes, which are surrounded by a lipid bilayer [43,47]. The limiting membrane of late endosomes plays an important role in ILV formation [48]. Among them, a protein complex—endosomal sorting complexes required for transport (ESCRT)— which exists on the surface of the limiting membrane, and raft proteins, such as phospholipids, cholesterol, caveolin, and tetraspanin, are actively involved in ILV formation [49]. These components help determine whether certain proteins should be transported by the ILV or be degraded by lysosomes. Exosomes contain various proteins derived from endosomes (ESCRTs, TSG101, etc.) and cell membranes (CD63, CD81, etc.) on their surface or inside their structure, and are involved in intracellular transport (Rab GTPase, etc.), in addition to various lipids derived from secretory cell membranes or endosomal membranes, such as cholesterol and sphingomyelin [50]. Among the different components, CD63, CD9, and CD81 are four-transmembrane proteins, called tetraspanins, that form a complex with other membrane proteins and functional molecules or serve as receptors, whereby a localized functional domain, called tetraspanin-enriched microdomain, is constructed on the membrane, leading to their involvement in cell–cell signal transduction, such as cell adhesion and fusion [51].

As exosomes are derived from the late endosome membrane, they contain proteins involved in intracellular vesicle transporters, such as annexin, low-molecular-weight G protein (the main component), and transmembrane proteins derived from the cell membrane, such as integrin, tetraspanin, membrane superficial protein, MFG-E8, raft-constituting proteins (such as caveolin and flotillin), cytoproteins that are taken up into vesicles when ILV is constricted (including actin, tubulin, various cytoplasmic enzymes, heat shock proteins), and proteins unique to each cell that secrete membrane vesicles [52,53]. Although knowledge of lipid components is limited compared to membrane vesicle proteins, the following findings have been reported: (1) lipid constituents of exosomes derived from reticulocytes and erythrocyte membranes are similar; (2) B-cell-derived exosomes contain lysobisphosphatidic acid, which is concentrated in the late endosome membrane; (3) phosphatidylserine, which is normally unevenly distributed in the inner lobe of the cell membrane, is present on the surface of exosomes; and (4) they are relatively rich in cholesterol [48,54,55,56]. Although exosomes have long been considered to play a role in intracellular mechanisms whereby these intracellular components are excreted, they have recently been recognized as an important delivery system for exchanging various proteins, nucleic acids, and lipids between secretory and target cells [57]. As immune-cell-derived exosomes have been demonstrated to contain antigen peptide–MHC complexes and various antigens, they have the potential to control various immune responses, such as the exchange of antigen information between immune cells and the regulation of immune cell activation [58]. In particular, exosomes secreted by B cells, T cells, and dendritic cells contain major histocompatibility complex (MHC) class I and II proteins that are required for antigen presentation [59,60]. Exosomes secreted from B cells are taken up by follicular dendritic cells and used for antigen presentation to T cells, inducing antigen-specific MHC class-restricted T cell responses [61,62]. In addition, the Fas ligand, localized in T-cell-derived exosomes, induces apoptosis in target cells via Fas and suppresses unnecessary cell responses [63]. Exosomes called retrosomes, secreted by small intestinal epithelial cells, have MHC class proteins bound to peptides derived from orally ingested antigens and are secreted to the basal side of the cells, inducing immune tolerance through movement into the blood [64]. mRNA and miRNAs derived from secretory cells have been confirmed to be abundantly present within exosomes. Accordingly, the nucleic acids within exosomes are confirmed to be involved in intercellular transmission and the regulation of gene expression. In this review, we opted to focus on the role of exosomes in the immune system and summarize recent reports and prospects for immune regulation and tolerance via miRNAs.

## 3. Regulation of Immune Response by Exosomes

The induction of T cell autoimmune tolerance, which is a central tolerance that occurs in the thymus, is essential for self-protection from autoreactive T cells [65]. Accordingly, the breakdown of this process causes autoimmune diseases. Central tolerance consists of three mechanisms: (1) positive selection to choose those that form a functional T-cell receptor (TCR) after reconstitution of the TCR, (2) negative selection to eliminate self-reactive T cells, and (3) differentiation of self-reactive T-cells into regulatory T-cells [66,67,68]. Positive and negative selections are induced by the contact of T cells with antigen-presenting cells in the thymus and the occurrence of antigen presentation [69]. Antigen-presenting cells in the thymus release exosomes and express MHC class I and II molecules [70]. Furthermore, the thymus exosomes can perform direct antigen presentation and are involved in T-cell differentiation in the thymus.

Dendritic cells (DCs), which are the most abundant secretors of exosomes in the immune system, act as a type of antigen-presenting cell by exhibiting fragmented antigen peptides from foreign substances, such as bacteria and viruses, that invade the body and are taken up by phagosomes following association with MHC class II molecules, on the surface for T cell activation [71,72]. On the other hand, intracellular antigen peptides are presented by MHC class I molecules. Thus, DC-derived exosomes contain both MHC class I and II on their surface and can activate T cells, even at a distant location from DCs, due to the interaction between T cell receptors, antigen-specific CD8+ cytotoxic T cells, and CD4+ helper T cells [73]. However, as the expression of the co-stimulatory molecules that are important for T cell activation is lower on the exosome surface than on the DC surface, the activation ability of exosomes is as low as 5 to 10% compared to that of DCs activated by direct contact with T cells [74]. In addition, exosomes not only perform such direct antigen-presenting functions by carrying an antigen peptide–MHC complex but also indirectly promote antigen presentation by incorporating exosomes into other antigen-presenting cells [75]. Furthermore, as exosomes contain various antigens derived from secretory cells, in addition to the antigen peptide–MHC complex, they have been shown to be transported to antigen-presenting cells [76].

Exosomes not only contain protein antigens, but also secretory-cell-derived mRNAs and non-coding RNAs, especially miRNAs [77,78]. As these RNAs are protected by the lipid bilayer membrane of exosomes, they are not degraded by the RNase and remain highly stable in blood and body fluids. Although some RNAs can be found in both exosomes and their secretory cells, some RNAs can only be detected in one of them. Therefore, the incorporation of specific RNAs into exosomes is assumed to be selectively regulated [79]. Thereafter, exosomes taken up by the target cell fuse with the endosome membrane to release encapsulated RNAs into the cytoplasm of the target cells. The released mRNA is then translated into protein; however, miRNAs suppress the translation of the target genes, enabling the exosomes to regulate gene expression in the target cells [80]. This mechanism is thought to be a means by which immune cells that encounter foreign invaders unite to counter their spread by horizontally transmitting their activated state to unencountered cells via RNAs [81]. As various immune-related molecules are encapsulated within exosomes, the immune responses are confirmed to be regulated by these molecules. For example, TNF family proteins, such as Fas ligand, TRAIL, and CD40 ligand, are present on the surface of exosomes derived from cytotoxic T cells and natural killer (NK) cells, leading to the apoptosis of target cells [82]. Similarly, some cancer cells are reported to release exosomes carrying Fas ligand, TRAIL, etc., inducing the apoptosis of immune cells to escape immunity [83]. In general, the TNF family proteins are produced as membrane-bound-type proteins and are cleaved by a membrane-type metalloprotease to form a soluble type, which has a relatively low activity of apoptosis induction compared with that of the membrane-bound type. Furthermore, exosome-encapsulated TNF family proteins are stable without being cleaved by membrane-type metalloproteases and have strong apoptosis-inducing activity by forming trimers through the membrane [84]. The transport of TNF family proteins through exosomes is involved in the development of various inflammatory and autoimmune diseases. For example, exosomes released from the synovial fibroblasts of patients with rheumatoid arthritis have high concentrations of membrane-bound TNF-α, which exacerbates the pathology of rheumatoid arthritis [85]. Human-cancer-cell-derived exosomes are also reported to have various immunosuppressive effects by suppressing interleukin 2 (IL)-dependent immune cell activation and causing the apoptosis of immune cells with NK activity by inducing cancer-specific cytotoxic T lymphocytes (CTL) through the amplification of regulatory T cells [86,87]. For example, when exosomes act on NK cells, they suppress the expression of the NKG2D receptor, which is responsible for the recognition mechanism of cancer cells and reduces cancer cytotoxicity [88]. However, very few reports have been published on the biological significance of the exosomes released by immune cells that are the main players in the anti-tumor immune response, such as T cells, NK cells, DC cells, and macrophages, and involvement in cancer cell proliferation, systemic circulation, and metastatic environment formation. When naïve CD3+ T cells were stimulated with CD3 and CD28 and cultured in an IL-2 medium, they were found to be rich in CD8+ T cells and contained exosomes in the culture supernatant [89]. miR-155 is essential for memory maintenance and the effector function of CD8+ T cells [90]. miR-155 is also known to release a variety of exosomes with different protein masses by activating CD4+ T cells. T lymphomas release a relatively large number of exosomes in culture, and T-cell-derived exosomes are considered to be immunologically significant. DC-derived exosomes, which are released abundantly during their maturation for the induction of antitumor immunity in T and B cells, and NK-derived exosomes, have damaging activity against various tumors [91].

When cancer-cell-derived exosomes are taken up by monocytes, the differentiation of monocytes into bone-marrow-derived immunosuppressive cells can suppress tumor immunity by inactivating immunocompetent cells and inducing regulatory T cells (Treg) through the secretion of various immunosuppressive molecules (such as IL-10), induced via the action of TGF-β and prostaglandin E2 containing the exosomes [92]. miR-17-92 suppresses Th1 cell function and amplification and inducible Treg (iTreg) cell differentiation by TGF-β, in which miR-17-92 is associated with the suppression of IL-10 production in iTreg cells [93]. Notably, miR-17-92 knockdown does not induce CD8+ T-cell memory. Through these molecular mechanisms, cancer cells are thought to suppress immune cell attack and promote cancer progression. However, miR-223 in exosomes produced by IL-4-activated TAM-like macrophages causes breast cancer cells to be more malignant [94]. Therefore, immune-cell-derived exosomes in a tumor environment may have properties that are elucidated during the malignant transformation of cancer. B-cell-derived exosomes are also taken up by CD169+ macrophages in the spleen and lymph nodes by systemic or subcutaneous administration [95]. Accordingly, target cells that are significantly taken up by exosomes may change depending on the type of parent cell released. In addition, exosomes are known to increase remarkably with age and are thought to be involved in tissue regression [96].

The physiological action of exosomes can only be clarified by revealing the mechanism of exosome secretion and the physiological phenomenon caused by its enhancement or inhibition [47]. To date, ultracentrifugation or PEG precipitation methods have been mainly used as general methods for the purification of exosomes; however, the resulting exosomes contain large amounts of impurities [97]. Thus, the experimental results might not be due to the action of the constituent molecules within the exosomes. Recently, novel methods for high-purity exosome isolation have been developed using affinity-based quantification, markedly aiding in the elucidation of the original physiological function [98]. Therefore, the development of new therapeutic methods using exosomes will progress as the function of exosomes in the immune system becomes clearer. For example, by inhibiting the function of cancer-cell-derived exosomes, the effects of anti-tumor immunity may be enhanced [87]. Alternatively, removing cancer-cell-derived exosomes from the blood of cancer patients may be an effective treatment. Conversely, manipulating functions, such as that of DCs, may stimulate the production of immunosuppressive exosomes, leading to therapeutic applications in the treatment of inflammatory and autoimmune diseases, where both miR-301a and miR-326 are amplified due to the proliferation of Th17 cells, which are IL-17-producing cells [99].

## 4. Exosomal miRNA-Mediated Molecular Pathways in Immune Response

Exosomes are suitable vesicles for in vivo cell-free transmission and intracellular delivery of proteins and nucleic acids, leading to a profound modulation of homeostasis [100]. As the biogenesis and secretion of exosomes and cargo sorting are stimulated by a hypoxic environment, the miRNAs encapsulated within the exosomes (i.e., miR-223, miR-23a, and miR-21) in hypoxia-induced inflammation can control the post-transcriptional regulation of inflammatory response pathways, including nuclear factor-kappa B (NF-κB), Toll-like receptor (TLR), and signal transducer and activator of transcription 3 (STAT3), via the polarization of macrophages, M1 and M2, ultimately influencing the inflammatory states [101,102,103,104]. The progression of inflammation under physiological hypoxia, in which insufficient oxygen levels for metabolic demand in the tissue initiate the hypoxia response, induces a series of metabolic changes, including an increase in glycolysis rate and oxygen consumption, promoting the formation of inflammatory mediators and innate and adaptive immunity through the oxygen-sensitive transcriptional regulator hypoxia-inducible factor (HIF) [105,106]. The HIF pathway that affects physiological homeostasis is activated by changing the metabolic state of immune cells, which reside within the tissue or are recruited from the oxygen-rich bloodstream via different oxygen-sensitive immune-related signaling pathways, ultimately modulating inflammatory products, such as cytokines, ROS, and nitric oxide [107].

During pathological hypoxia, where chronic insufficient oxygen supply for the metabolic needs of the tissue causes immune cell activation through inflammation, oxidative stress, and cell death, the biogenesis of exosomes is increased [108]. In fact, the number of exosomes increases, but their size remains the same. The hypoxic state can stimulate the rearrangement of the cytoskeleton and extracellular matrix of exosomes, leading to the reprogramming of recipient cells [109]. In response to hypoxia, HIF-1 participates in the sorting of miRNAs, such as miR-23a, into exosomes and the release of exosomes into the extracellular space via transcriptional regulation of hypoxia-responsive genes via the binding of complexes comprising HIF1α, HIF1β, and coactivator p300/CBP within hypoxia-responsive elements in the nuclei [110]. In contrast, exosomes can induce the activation of HIF-1a, which mediates the release of cytokines, indicating a synergistic role between exosomes and HIF1α in regulating the process of immune inflammation. Hypoxia-driven exosome secretion is modulated by the activation of the STAT3 pathway due to the association of HIF-1a with a family of small GTPases, Rab GTPases, through cytoskeletal and sub-membranous actin rearrangement [111].

Oxidative stress, which is the imbalance between free radicals and antioxidants under hypoxia, affects miRNA sorting into exosomes by changing the composition of membrane lipids and the secretion of exosomes [112]. Exosomes derived from mesenchymal stem cells (MSCs) can alleviate oxidative stress via activation of the PI3K/Akt signaling pathway to enhance cellular survival and reduce inflammation [113].

The HIF-1a and HIF-1b levels are regulated by Th1 (IFN-γ-and LPS) and Th2 (IL-4 and IL-13) cytokines released from M1 and M2 macrophages, respectively [114]. miR-29 and miR-146 strongly suppress IFN-γ-production by Th1 cells, and elevated miR-146a levels reduce TLR9-mediated IL-12 production by DC via CD11b signaling and suppress CTL cross priming [115,116]. Macrophage polarization requires the reprogramming of intracellular metabolism and exhibits unique characteristics of glucose metabolism, which are induced by differences in inflammatory dynamics and duration. In the activation of M1 macrophages, the anaerobic glycolysis pathway is induced, owing to the need for rapid and large amounts of energy that quickly produce ATP to manage the persistent inflammation, manifested as increased glucose consumption and lactate release [117]. Although the tricarboxylic acid cycle (TCA) is suppressed, the TCA intermediate, succinate, is increased, which induces IL-1β production through the stabilization of HIF-1α [118]. On the other hand, M2 macrophages mainly represent the oxidative glucose metabolism pathway, in which large amounts of ATP are efficiently produced by oxidative phosphorylation [119]. Furthermore, arginase 1 (Arg1) production by M2 macrophages induced by HIF-2α is mainly involved in cell proliferation, inflammation, and tissue remodeling via the migration, invasion, and regulation of chemotactic receptor expression. The promotion of M2 macrophages from M1 macrophages is regulated by the targeting of the PBX/Knotted 1 Homeobox 1 (Pknox1) or NLR family, pyrin domain containing3 (NLRP3) by miR-223, leading to the prevention of activation of inflammation or inflammasome through the decrease in IFN-g-and IL-6 and increase in IL-10 [120]. During the differentiation of monocytes to macrophages, the inhibitory effect of IkappaB kinase alpha (IKKα) is alleviated, leading to the promotion of inflammation via activation of the nuclear factor-kappaB (NF-κB) pathway [121]. Furthermore, the suppression of NLRP3 or the production of excess ROS by targeting the chemokine CXC receptor 4 (CXCR4) or the peptidylprolyl isomerase F (PPIF), which is the gatekeeper of the mitochondrial permeability transition pore, with miR-23a, respectively, induced anti-inflammatory effects [122]. The targeting of programmed cell death 4 (PDCD4)/caspase-3 and NFκB-TNF-α-TLR pathways by HIF-1α-induced miR-21 upregulation induces anti-apoptotic and anti-inflammatory effects, respectively, by suppressing the secretion of inflammatory cytokines, including IL-6, IL-1β, and TNF-α [111,123].

Under hypoxic conditions, the polarization of CD4+ helper T cells is suppressed to prevent excess activation in the immune system, leading to inhibition of the adaptive immune system and induction of innate immune cells [124]. Furthermore, differentiation of monocytes into macrophages is promoted by the decreased expression levels of miR-223, miR-15a, and miR-16, leading to the polarization of M1 macrophages; this is because of the enhanced secretion of IL-6 induced by the inhibition of the JAK/STAT signaling pathway with miR-155 expression, and the reduced PI3K/Akt-mediated IL-10 production by induction of PTEN with the downregulation of miR-3473b on the IFN-g-priming effect [125,126,127]. M2 macrophage polarization is suppressed by the inhibition of IL-4-mediated JAK1/STAT6 or IRF4/PPARg pathways with miR-23a or miR-27 expression, respectively (Figure 2) [128]. However, the polarization of M2 macrophages is promoted by the reduction in IFN-g-and IL-6 and induction of IL-10 through the targeting of Pknox1 with miR-223, and the decreased production of PGE2 through STAT3 targeting with miR-21 [111,129,130,131,132]. M2 macrophage polarization is also suppressed by the indirect regulation of the TNF-α signaling pathways, such as MyD88 and TRAF6 with miR-146a, and the inhibition of IL-6 through TLR4 signaling; the association of HIF-1α with the promoter region of TLR4 under hypoxia enhances the response of inflammation in macrophages with miR-let-7b [111,133]. Additionally, the polarization of M1 to M2 macrophages in vivo and in vitro is regulated by miR-182 within exosomes from MSCs by targeting TLR4 [134]. Further, CD47, a “don’t eat me” signal can modulate the ability of macrophages as a major phagocytic barrier through an interaction of signal regulatory protein α (SIRPα) on nature-derived exosomes [135]. In addition, syntenin-1 regulates the sorting of Toll-like receptor 7 (TLR7), which contributes to autoimmune diseases, such as systemic lupus erythematosus via recognition of self-RNA, into intralumenal vesicles of multivesicular bodies [136]. These exosomal miRNAs would play critical roles on the modulation of immune response. Therefore, it could be applied to new drug delivery systems (DDS) of miRNAs for various diseases.

## 5. Conclusions

The exosomes secreted from DCs and macrophages as antigen-presenting cells play critical roles in the immune system to activate T cells via the presentation of fragmented antigen peptides from foreign substances, such as bacteria and viruses, invading the body by MHC. miRNAs encapsulated in exosomes modulate the expression of target genes in signaling pathways, such as NF-κB, STAT1, STAT3, and PI3K/Akt, in the immune response via post-transcriptional regulation. This is a novel strategy for various autoimmune diseases, as exosomal miRNAs can be used as cell-free DDS.

## Figures and Tables

**Figure 1 ijms-23-05658-f001:**
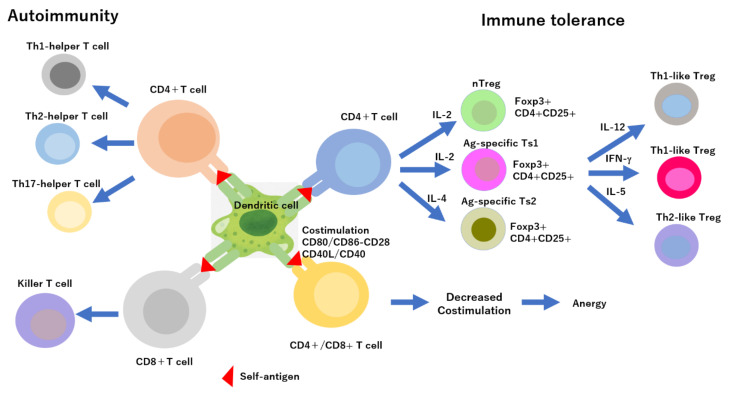
Autoimmunity and immune tolerance induction in dendritic cells. Through the interaction of dendritic cells (DCs) with T cells via major histocompatibility (MHC)-class I or -class II and co-stimulatory receptors and cytokine secretion levels, pro-inflammatory DCs presenting self-antigen can prime CD8+ helper and CD4+ killer T cells, promoting autoimmune diseases and transplant rejection in autoimmunity. However, the DCs in immune tolerance express specific cytokines and receptors, inducing regulatory T cells or anergy. The expanding natural T regulatory cells (nTregs) induce polyclonal activation with IL-2.

**Figure 2 ijms-23-05658-f002:**
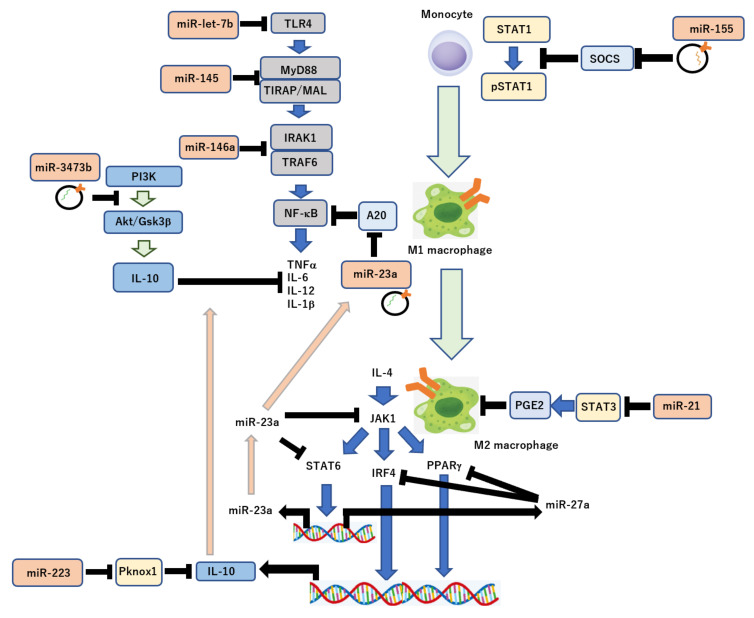
Modulation of immune response by miRNAs. Polarization of macrophages, M1 to M2, differentiated from monocytes, regulated by some molecular pathways, including NF-kB, STAT1, STAT3, and PI3K/Akt pathways, via post-transcriptional regulation by miRNAs, such as miR-155, miR-21, miR-let-7b, miR-145, miR-146a, miR-3473b, miR-23a, and miR-223.

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
