# Peer review of "Immune Modulation Using Extracellular Vesicles Encapsulated with MicroRNAs as Novel Drug Delivery Systems"

_ijms, 2022, doi:10.3390/ijms23105658_

Round 1

Reviewer 1 Report

 Matsuzaka and Yashiro have complied review on, ‘Immune modulation using extracellular vesicles encapsulated with 2 microRNAs as novel drug delivery systems’ very elegantly. The review manuscript is well written, and Figures are adequate. There is minor suggestion: 

Tetraspanins are widely used as exosome markers in the extracellular vesicular field, However, new evidence has emerged that many surface markers such as CD47, SIRPα and syntenin-1  (PMID: 34108659; PMID: 35235473; PMID: 35004730; PMID: 34841476; PMID: 29368647; PMID: 35485671; PMID: 35325005) can be useful for drug delivery systems. Authors should discuss prospects of their role Modulation of immune response by miRNAs in the review. 

Author Response

Matsuzaka and Yashiro have complied review on, ‘Immune modulation using extracellular vesicles encapsulated with 2 microRNAs as novel drug delivery systems’ very elegantly. The review manuscript is well written, and Figures are adequate. There is minor suggestion: 

Tetraspanins are widely used as exosome markers in the extracellular vesicular field, However, new evidence has emerged that many surface markers such as CD47, SIRPα and  (PMID: 34108659; PMID: 35235473; PMID: 35004730; PMID: 34841476; PMID: 29368647; PMID: 35485671; PMID: 35325005) can be useful for drug delivery systems. Authors should discuss prospects of their role Modulation of immune response by miRNAs in the review. 

According to the reviewer’s comments, we added the discussion about the modulation of immune response by miRNAs, on line 413-420, in page 12-13.

Reviewer 2 Report

In this review, the author gives an overview of the immune modulation mediated by miRNA contained in small extracellular vesicles.

Introduction

The introduction is well structure, complete and not too long (not redundancy). Minor comment, "γ" is missing in line 51.

Core of the review

Similarly to the introduction, the core of the manuscript  is well written and clear. Despite this, few changes would greatly improve it.

Line 107. The term "exosome" should be explained before. To the reader, the difference between membrane vesicles and exosomes should be clear.

Line 108. Fraction distribution requires density gradient, this should be stated.

Line 111. An endocytotic independent fusion also exist.

Line 116 - 117. Not clear. MVB are not exosomes, but contain exosomes.

Line 131 - Instead of "Among them", I would suggest "Among the different components". Also, depending on the type of parental cells, CD9 is also a known exosome component.

Section 3 and 4 would benefit each of a summarizing table. 

It is important in the tables and within the revision to state exactly when the references are about exosomes or other extracellular vesicles.

Author Response

The introduction is well structure, complete and not too long (not redundancy). Minor comment, "γ" is missing in line 51.

According to the reviewer’s comments, we corrected IFN-gamma, on line 51, in page 2.

Core of the review

Similarly to the introduction, the core of the manuscript is well written and clear. Despite this, few changes would greatly improve it.

Line 107. The term "exosome" should be explained before. To the reader, the difference between membrane vesicles and exosomes should be clear.

According to the reviewer’s comments, we added the explanation of the exosomes, on line 107-109, in page 3.

Line 108. Fraction distribution requires density gradient, this should be stated.

According to the reviewer’s comments, we corrected the fraction distribution as 1.12 to 1.19 g/mL, on line 109, in page 3.

Line 111. An endocytotic independent fusion also exist.

According to the reviewer’s comments, we corrected this sentence as “clathrin-, caveolin-, and lipid raft-mediated endocytosis, phagocytosis, micropinocytosis, or membrane fusion” on line 113 - 114, in page 4.

Line 116 - 117. Not clear. MVB are not exosomes, but contain exosomes.

According to the reviewer’s comments, we corrected this sentence as “ILVs are exocytosed via the fusion of MVBs containing some ILVs with the cell membrane as exosomes which are surrounded by a lipid bilayer [43, 47] ” on line 119 - 121, in page 4.

Line 131 - Instead of "Among them", I would suggest "Among the different components". Also, depending on the type of parental cells, CD9 is also a known exosome component.

According to the reviewer’s comments, we corrected this sentence as “Among the different components, CD63, CD9, and CD81 are four-transmembrane proteins, “ on line 132 - 133, in page 4.